# Patient lead users experience of the COVID-19 pandemic: a qualitative interview study

Hanna Jansson ,[1] Terese Stenfors ,[2] Sara Riggare ,[3] Henna Hasson ,[1,4] Maria Reinius [1]

¹Department of Learning, Informatics, Management and Ethics, Medical Management Centre, Karolinska Institutet, Stockholm, Sweden
²Department of Learning, Informatics, Management and Ethics, Division of Learning, Karolinska Institutet, Stockholm, Sweden
³Healthcare Sciences and e-Health, Department of Women's and Children's Health, Uppsala University, Uppsala, Sweden
⁴Unit for Implementation and Evaluation, Center for Epidemiology and Community Medicine (CES), Region Stockholm, Stockholm, Sweden

**Correspondence to**
Dr Hanna Jansson;
hanna.jansson@ki.se

## ABSTRACT

**Objectives** Patient lead users can be defined as patients or relatives who use their knowledge and experience to improve their own or a relative's care situation and/or the healthcare system, and who are active beyond what is usually expected. The objective of this study is to explore patient lead users' experiences and engagement during the early COVID-19 pandemic.

**Design** Qualitative in-depth interviews with a cross-sectional time horizon.

**Setting** The early COVID-19 pandemic in Sweden, from 1 June through 14 September, 2020.

**Participants** A total of 10 patient lead users were recruited from the Swedish patient lead users (*spetspatient*) network. All participants were living with different long-term conditions and matched the definition of being patient lead users.

**Results** We found that during the early pandemic, patient lead users experienced that they no longer knew how to best manage their own health and care situations. On an individual level, they described an initial lack of knowledge, new routines, including a change in their health and an experience of people without a disease being in the same situation as them, for a while. On a systemic level, they described a fear of imminent unmet-care backlogs and decreased opportunities for sharing patient perspectives in care organisation, but also described increased networking.

**Conclusions** Patient lead users can be seen as an emerging community of practice, and as such could be a valuable resource as a complementary communication channel for an improved health system. The health systems were not able to fully acknowledge and engage with the resource of patient lead users during the pandemic.

## INTRODUCTION

Many individuals with long-term illness become expert patients[1] and often take the main responsibility for management of their condition.[2 3] They engage in active self-care[4] and are often well informed regarding their condition.[5] Sometimes, when formal healthcare is unable to respond to patients' needs, individuals with long-term illnesses create or develop new solutions, or tweak already existing solutions to better fit their own

## STRENGTHS AND LIMITATIONS OF THIS STUDY

⇒ The qualitative and inductive approach contribute to an in-depth and rich exploration of patient lead users' experiences during the COVID-19 pandemic.
⇒ A sample consisting of respondents with different chronic conditions and life situations.
⇒ A patient lead user was part of the research team, which can contribute to improving the relevance of research to patients.
⇒ The relatively small sample might affect the reliability of the study.

needs.[3 6 7] Inspired by von Hippel's framework on lead users,[8] we propose that these patients, active beyond what is usually expected, be called patient lead users.[9 10] A patient lead user network was established around 2016–2017 as a patient-driven initiative with the aim of supporting and stimulating the implementation of integrated healthcare systems from a patient (and informal caregiver) perspective. The network works across diagnoses in Sweden and internationally.

Research on patient lead users has found that active and engaged patients can go from information seekers to innovators.[11] Thus, patient lead users are individuals accustomed to informing themselves and managing their own well-being. Their actions can be understood from a recent study proposing a framework for the different roles that patient lead users can display.[12] The framework describes how patient lead users engage in multiple behaviours and activities that target improvements in their own health and well-being and improvement of the health system and society at large. A related study[11] indicates that patient lead users are characterised by a desire to do more than is often possible, given the roles and activities established in health systems. Furthermore, patient lead users can develop behaviours and competencies that may serve to inspire other patients and their informal caregivers.

During the spring of 2020, the COVID-19 outbreak spread rapidly around the world, with severe effects on health systems and societies globally. In Sweden, as in many other countries, the COVID-19 pandemic occurred in waves of increased transmission. Following the first wave, which started in March 2020, a period of lower transmission rates occurred between mid-July and October, before the second and third waves occurred.[13] This caused tremendous uncertainty and worry for people who got infected by COVID-19 and other people in need of care, and a recent study showed that patients in general experienced anxiety and hesitation regarding care contacts, especially for routine care and elective procedures and surgeries, due to fear of infection.[14]

In the acute situation of the COVID-19 pandemic, the conditions for healthcare changed drastically. In the Swedish healthcare system, clinical wards for infectious diseases and intensive care units quickly needed to expand their capacity in terms of rooms, healthcare staff and medical equipment. In March 2020, the number of people infected with COVID-19 and needing inpatient care increased quickly in Sweden.[15] The capacity in clinics for infectious diseases and intensive care had to be expanded within weeks. Healthcare wards and staff usually working with infectious diseases focused only on providing care for persons with COVID-19, and staff from other wards were relocated to work on these dedicated COVID-19 wards.[15] Since healthcare resources needed to be relocated both within regions and nationally, the access to other types of healthcare was decreased. Retrospective studies of healthcare provided during the pandemic show that planned care was cancelled, care visits were converted from physical to digital appointments and people in need of care waited to book appointments in fear of burdening healthcare in this challenging situation.[15] Despite the many retrospective studies conducted, the experiences and actions of patient lead users have experienced during this period of societal and healthcare crisis remain unknown. The aim of the current study was therefore to explore patient lead users' experiences during the early COVID-19 pandemic. This included a desire to better understand how patient lead users can function as a possible resource for health systems. More specifically, the study addresses the following research questions:

1. How did patient lead users in Sweden experience their situation during the early pandemic?
2. What behaviours and activities did patient lead users in Sweden engage in during that time?

## METHODS
### Design and recruitment
Qualitative research was conducted following an interview strategy over a cross-sectional time horizon. Since the study aimed to explore the experiences of patient lead users, non-probability purposive sampling was used to select participants for the study. Participants were recruited from the Swedish patient lead users (*spetspatient*) network. E-mail invitations were sent by two patient lead users that were coresearchers in the project. Following the arguments proposed by Varpio *et al*,[16] we aimed for a sample that is adequate (of sufficient size to allow transferability to other contexts), appropriate (with data that can answer the research questions) and aligned (with research questions and methodological choices). Reminders were sent if no response was received. In total, 13 patient lead users were invited, and reminders were sent to those who did not respond. At the end, 10 agreed to be interviewed. Personal experience of having been sick with COVID-19 was not an inclusion criterion.

### Data collection
Semistructured interviews were conducted based on an interview guide. The first draft of the interview guide was created by TS, SR and HH and later jointly developed by the research team, to guide the interviews with open-ended key questions while leaving room for flexibility and for the participants to elaborate on their answers. The interviews started with a short introduction to the study and informed consent was collected. The participants were then asked to introduce themselves, including their disease history and role as a patient lead user. After this, the questions focused on two key areas in relation to the COVID-19 pandemic: (1) challenges in health and care routines, and (2) opportunities through new behaviours and activities. Follow-up questions like 'can you tell me more about that?' were asked to probe deeper into key areas.

Individual single interviews were conducted by a female research assistant trained in qualitative methods and interview techniques during the early pandemic, from 1 June to 14 September 2020. The interviews, conducted using Microsoft Teams, lasted 35–86 min (average 60 min), and were audio recorded (no video) and transcribed verbatim.

### Data analysis
The data from the interviews were analysed using reflexive thematic analysis.[17 18] To meet the aim of the study, a semantic and inductive approach was chosen, allowing the data to drive the analysis and theme development.[18] The analysis was based on the Braun and Clarke six-phase guide to performing thematic analysis,[17] including reflexive dialogue throughout the analysis.[17 18] Microsoft Word and Excel were used to manage the data.

All authors read the transcripts to familiarise themselves with the data. Thereafter, two authors (HJ and TS) read all interview transcripts to note initial ideas. One author (TS) coded the data by looking for features that corresponded to the aim and collated all codes into potential themes. One author (HJ) reviewed the themes in relation to the coded extracts and the entire data set. The same two authors continued to refine and name the themes. As a final step in the data analysis, all authors contributed

| Table 1 | Summary of themes and subthemes |
|---|---|
| **Themes** | **Subthemes** |
| Individual level | Initial lack of knowledge: 'Am I in a high-risk group?' |
| | New routines and changes in individual health |
| | 'Everyone else was in the same situation as me, for a while' |
| System level | Concern for a backlog of unmet healthcare needs |
| | Patient involvement: a supplemental activity and first to be cut |
| | Increased networking among patients |

to the manuscript, finalising the analysis relating to the study's aim and previous studies.

## Patient and public involvement

If a study aims to improve the relevance of research to patients, this must also be reflected in the research process used.[19] Therefore, the research team for this study included authors with formal experience as researchers, healthcare workers and with lived experience as patient lead users. A patient lead user with lived experience of long-term disease has coauthored this article, been part of the research team and has contributed in all parts of the research process.

## RESULTS

Ten individuals (nine women, one man), all adults and identifying as patient lead users, were interviewed. The participant group entailed persons living with neurological, rheumatological, muscular, haematological, pulmonary and metabolic long-term conditions as well as experiences of cancer and paraplegy. The data analysis generated themes on two levels, the individual and system levels, where individual-level themes describe experiences related to oneself, and system-level themes describe experiences of actions taken on behalf of other patients and the healthcare system as well (table 1). The themes are described below and illustrated with quotes from the interviews.

## On an individual level

### Initial lack of knowledge: 'Am I in a high-risk group?'

At the time of the interview, participants were unsure whether they were considered, or considered themselves to be, in a high-risk group. They had searched for information and/or discussed this with their health professionals, and some received clear answers, others not. Participants mentioned being told that authorities could not use data from other countries, they had to wait for Swedish data. This information also changed often, which created increased stress and worry for some and affected their daily lives, as they did not know whether they should go to work or stay home.

> It has been very difficult that the information has been so varied from different sources (Respondent A)

Another participant experienced receiving different information from different sources.

> Some authorities lumped us all together, just saying, we are a high-risk group. But then… the leading MS neurologists in Sweden have said no, you are not. (Respondent B)

Respondents reported identifying and assessing the available information themselves.

> There is no aggregated information. […] And you had to try to find your way forward yourself, and… find information and make judgements based on what you hear. (Respondent C)

One respondent described how reports from the media and public authorities made them think that they belonged to a high-risk group that should self-isolate. The respondent described a sense of guilt among individuals in the group, as if belonging to a high-risk group were their fault.

### New routines and changes in individual health

Due to the restrictions imposed by public authorities, the respondents spent more time at home and worked from home, which led to new routines. The new routines were in many cases beneficial; one respondent explained the relief of not having to wear uncomfortable clothing, and another respondent described how they could rest their joints much more during this period. Perhaps surprisingly, many respondents described an improvement in their health and well-being. One respondent shared that life was easier now, when they could find time to rest and get energy, and another pointed to increased opportunities for recovery.

> My body has really benefited from it, physically, although mentally it has become a concern. (Respondent D)

But for some, the situation led to a lack of physical activity that was detrimental to their health. One respondent specifically pointed at the scientific evidence for the importance of physical activity for her disease, something that was much more difficult during the pandemic, when all group exercise classes at gyms were cancelled. Increased stress and worry in general were considered difficult by several.

> You stay more isolated at home, and then it perhaps becomes more difficult to be healthy. (Respondent E)

One respondent saw the decline in her health as an important eye-opener and a reminder of her vulnerable health status.

The coronavirus has turned the whole 'self-care' practice upside down […], made me sicker since stress, mental-health issues, and other external, or surrounding (factors) have affected my ability to have a functioning self-care regimen. (Respondent F)

Another respondent reflected on her close contact with other patients who were extremely unwell. She also knew many who died, if not directly from the virus, then indirectly due to delayed healthcare interventions, and all this affected her own health as well.

People have died and board members (in my patient organisation) have died, and… And everything is put on hold, right. So that it has affected me a lot mentally. For a while, last summer, I was completely incapacitated. I could not manage anything. (Respondent G)

### 'Everyone else was in the same situation as me, for a while'

Many participants had already experienced major changes in their life situation, related to their chronic condition. During the pandemic, when public restrictions changed life for the majority of people, some respondents suggested that their lives had not changed much. The difference, respondents suggested, was much bigger for 'others' who did not have similar prior experience. The knowledge that everyone was in a similar situation, with restricted opportunities for travel and socialising, for example, was felt to be helpful. One respondent suggested that perhaps now, after others experienced the limitations of life with a chronic condition, understanding could increase in the future.

Out of everyone in my group at work, I suffer the least from this…. I can feel like I haven't fallen so far… My everyday life is strangely limited anyway, so there is no difference. (Respondent B)

However, one respondent experienced that, as soon as the pandemic situation improved, there was a lack of understanding for her continued restrictiveness in travelling. She was asked to participate in a panel debate, and although the event was digital, the panel was invited to meet onsite. After experiencing the equalising effects of the early pandemic, when no one could travel, the respondent felt more excluded than ever.

### On a system level
#### Concern for a backlog of unmet healthcare needs

Some respondents mentioned that care appointments had been cancelled or rescheduled. Even if one respondent was happy that care was postponed, since it felt safer to avoid the hospital for the time being, there was a general concern about the long-term consequences of cancelled care. Several respondents mentioned their concern for an accumulating backlog of unmet care in general.

But for the target group in general, it is still a lockdown in many places, and when you start scheduling (non-acute) surgery again, the difficult patients are really prioritised. For obvious reasons. But there is a large backlog of unmet care that must be handled. (Respondent F)

#### Patient involvement: a supplemental activity and first to be cut

Some participants felt that people with a chronic condition, who need continuous care combined with a concern for secondary diseases and severe infections, had been somewhat forgotten about. At the same time, there was a hope that the pandemic might have created an increased awareness among people regarding what it is like to live in constant fear of sickness and worry. Some respondents expressed that their ability to influence their care organisations at large had diminished during the pandemic. Their experience was that prepandemic patient involvement was now neglected or forgotten about.

The patient perspective has fallen through the cracks. (Respondent H)

But despite feeling that their ability to currently influence the healthcare system had diminished, many felt this was reasonable given the extreme situation. The experience that patient involvement was not prioritised during the crisis was considered a sign that patient involvement is not yet an integrated part of the system, but rather a volunteer and supplemental activity. It was suggested that patient involvement would have been beneficial even given the extreme circumstances.

It has not been requested now, during the crisis situation. Instead, it is pushed aside. […] I think that is a sign that this is not… built into the system… structurally, properly, with patient involvement. Rather, it is something that happens more on a voluntary basis, off-handedly, and as a box-checking exercise. (Respondent C)

Others saw opportunities here, considering the time after an extraordinary situation to be propitious for changing the system. Even during the pandemic, some respondents were asked to share their thoughts. Some mentioned patient councils at different hospitals.

It feels a bit like there is, or was, a time before the pandemic, and there is sort of a time after. And there is, like, room for change, and greater investments in healthcare. So, it is a… current issue. Many politicians sort of listen to what we have to say right now. (Respondent E)

#### Increased networking among patients

All respondents were actively involved in online communities. These included Facebook groups and social online meetups, where the respondents engaged in discussions with other patients to make sense of the new situation and support each other. In these online open fora, the participants were actively engaged, intending to contribute to

a nuanced discussion based on information the most reliable information.

> You find others with the same diagnosis. Because if you only had contact with healthcare, then you would go completely crazy. After all, healthcare cannot tell a person with a chronic diagnosis how to live with the diagnosis every day. [...] And without the patient lead user movement, I would probably have gone crazy, actually. Like... I would have been so frustrated and unable to find a way to channel everything I am carrying around inside. (Respondent B)

Other examples mentioned were digital activities with educational purposes, such as webinars to support others engaged in self-care and lectures with invited experts. Many participants described how they had engaged professionals, such as physicians, psychologists or dieticians, in these networks for Q&A sessions, advice and support.

> We can sit and speculate to death here (in our networks), but we still must listen to those who have somewhat more to contribute. So, we have tried to connect with what is sort of reasonable and sensible. (Respondent B)

For some, being active in these networks was described as part of their professional responsibility. Others did not stop at networking; they also engaged in more strategic lobbying.

> But I think that you—well, that I have some kind of drive or curiosity to improve. I think that the healthcare situation has been really bad. And you want it to be equal for everyone, and... no matter where you live. (Respondent G)

Some respondents described a decrease in activism opportunities, since all big political events in Sweden were cancelled due to the pandemic, and a subsequent need to find new channels for this work.

> Almedalen week, Järva, Pride (large annual political events) are all cancelled, all those things we usually attend. (Respondent A)

One of the respondents had coinitiated a new network in collaboration with a pharmaceutical company. The aim of the network is to create a strategy for strengthening patient voices during future situations like the COVID-19 pandemic.

## DISCUSSION
### Principal findings

During the early COVID-19 pandemic, neither evidence-based knowledge, nor information about its effects on individuals with specific illnesses, was available about the virus. Stern et al[20] have previously described a quality deficit of web-based information during the early phase of the pandemic. Patient lead users, accustomed to being informed about their condition, suddenly found themselves in an information vacuum, where very little was known about possible consequences for their health and well-being. This study shows how patient lead users experienced stress due to this lack of information. Similar experiences are described in a study on public health and risk communication during the COVID-19 pandemic.[21] Porat et al describe an overabundance of information, some accurate and some not, leading to increased risk for depression and anxiety.[21] In response, they propose guidelines to support well-being, of which one is to apply a 'bottom-up approach to communication' by engaging different stakeholders in coproduction. This could relate to the result of our study, which shows how patient lead users tried to search and compile information themselves, made the information available to others and took a leading role in understanding the situation.

Furthermore, healthcare was overburdened during the early COVID-19 pandemic, and the patient lead users described decreased opportunities for providing patient perspectives on a healthcare-system level. When prioritisations had to be made, and care was postponed or cancelled, the ordinary channels for patient involvement were no longer available. The patient lead users in this study also described fearing an imminent backlog of unmet care, but respondents nevertheless also described this period, as healthcare must adapt to a new situation, as propitious for change. Early research has also discussed the opportunity for change that can come with the lessons learnt after a great challenge. Duek and Fliss[22] describe how, in parallel with challenges, there were opportunities for change and improvement, for innovations and creative solutions, and Abrams et al[23] describe how to reconsider what is meant by quality for the healthcare system postcovid. Some early studies have already considered patient involvement and shared decision-making during the COVID-19 pandemic: Abrams et al[24] discuss COVID-19-related challenges for shared decision-making concerning health service reallocation, communication of uncertainty, social media influence and re-evaluation of assumptions guiding today's practices. Decision-making was also addressed in an editorial by Richards and Scowcroft,[25] who pointed out that patients were not partners, and were not consulted as experts in the necessarily fast decision-making of the pandemic. Our findings confirm this and illustrate how patient lead users perceived decreased opportunities for patient engagement during the first phase of the COVID-19 pandemic.

The result of this study also describes increased networking and community building. Richards and Scowcroft[25] describe how many patient and civil society advocacy groups chose to take an active role in providing information and support for their communities but also lobbied for patient representation in policy-making during the pandemic.[25–27] The UK national quality standard on community engagement[28] highlights three types of community engagement in the context of the COVID-19 pandemic: community engagement through

the formation of new networks, through developing new skills within existing organisations, and through individual efforts by people with specific local knowledge from the experience of multiple challenges. The result of this study shows how Swedish patient lead users have engaged in all three types of engagement during the pandemic. The respondents described how their peers became even more important during the pandemic. New networks were formed, patient organisations adapted to the unique circumstances and developed their services, and patient lead users acted as voluntary moderators on digital platforms to support their peers. The actions described by this study can be seen as signs of an emerging community of practice,[29 30] where patients lead users share the knowledge and experiences gained in their practical activities as patient lead users. Patient lead users build a sense of belonging through empowerment and growth, as well as providing support for others ('without the patient lead user movement, I would probably have gone crazy') in the patient community.[31] Looking at the patient lead users' activities as part of a community of practice could allow further studies to shed more light on patient lead users' actions, providing guidance for their development and use as a potential resource for the health system.

### Strengths and weaknesses of the study

The interviews with patient lead users about their experiences provided rich data about their activities and behaviours during the current COVID-19 pandemic. Although all respondents were from Sweden, they differed regarding diagnoses, providing variation in collected data, with experiences from different perspectives. A sample with a majority of female participants could affect the generalisability of the result but gender was not a sampling criterion in this study. The term patient lead user is still new, and recruitment based on this criterion was not easy. At the end of the interview, the participants were asked if they knew other patient lead users interested in the study, to allow for an additional snowball sampling approach. Yet, no further respondents were identified in this way.

### Implications and future research

This study aimed to explore patient lead users' experiences during the first phase of the COVID-19 pandemic, and the result illustrates the potential of these patients even under such uncertain and demanding conditions as a pandemic. The patient lead users possess a great deal of knowledge and their firm ambition to contribute to better care remains. Healthcare staff and policymakers may benefit from the work of individual patient lead users or their networks. Not least during a crisis, patient lead users can be a resource, for example, by spreading information to others in similar situations.

This study provided a snapshot of patient lead users' experiences during the first phase of the COVID-19 pandemic. We acknowledge that the societal response or action during the pandemic varied greatly in different countries, hence, this needs to be considered as an aspect of the transferability of our findings. Still, future research could consider longitudinal studies following the development of the situation and return to ordinary life, or at least life without an ongoing pandemic. Another interesting perspective would be to explore the experiences of healthcare staff and policymakers and their perceptions of patient involvement during the pandemic. For example, could patients have been involved in another way, and do healthcare staff and policymakers have any recommendations for future similar situations?

## CONCLUSIONS

This study provides an insight into the experiences of a number of patient lead users in Sweden during the first phase of the current COVID-19 pandemic. Our findings suggest that patient lead users can be seen as an emerging community of practice and as such could be a valuable resource as a complementary communication channel for an improved health system. However, the result shows that respondents experienced that neither individual care of people with chronic conditions, nor patient involvement, were prioritised during the pandemic. The health systems were considered unable to fully acknowledge and use the resource of patient lead users.

**Acknowledgements** First, we would like to thank the participants in this study. The authors also would like to thank Kim Nordlund, for participating as coresearcher in the beginning of the research process, and research assistant Johanna Stjärnfeldt, for her work with scheduling, performing and transcribing the interviews.

**Contributors** HH and SR initiated the study. TS led the data collection. SR identified and invited participants. TS and HJ analysed the data, and TS developed preliminary themes that were reviewed by HJ. HH drafted the background, TS drafted the result and HJ drafted the method and discussion sections of the manuscript. All authors revised, edited and approved the final manuscript. MR was responsible for the overall content as the guarantor. HJ, as the corresponding author, attests that all listed authors meet authorship criteria and that no others meeting the criteria have been omitted.

**Funding** This work was supported by the Swedish Research Council for Health, Working Life and Welfare (FORTE) grant number 2018-01472.

**Competing interests** None declared.

**Patient and public involvement** Patients and/or the public were involved in the design, or conduct, or reporting or dissemination plans of this research. Refer to the Methods section for further details.

**Patient consent for publication** Not required.

**Ethics approval** This study involves human participants and was approved by Swedish Ethical Review Authority, 2019-03849 with amendment 2020-01741. Participants gave informed consent to participate in the study before taking part.

**Provenance and peer review** Not commissioned; externally peer reviewed.

**Data availability statement** No data are available. No additional data available.

**ORCID iDs**
Hanna Jansson http://orcid.org/0000-0002-1615-0463
Terese Stenfors http://orcid.org/0000-0002-0854-8631

Sara Riggare http://orcid.org/0000-0002-2256-7310
Henna Hasson http://orcid.org/0000-0002-3827-6841
Maria Reinius http://orcid.org/0000-0003-0864-8701

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
