## [Reviewer comments · BMJ Open]

ARTICLE DETAILS

TITLE (PROVISIONAL)	Patient lead users experience of the Covid-19 pandemic, a qualitative interview study
AUTHORS	Jansson, Hanna; Stenfors, Terese; Riggare, Sara; Hasson, Henna; Reinius, Maria

VERSION 1 – REVIEW

REVIEWER	Lewis, Dana OpenAPS, Seattle
REVIEW RETURNED	12-Jan-2022

GENERAL COMMENTS	I appreciated the opportunity to review this manuscript focused on exploring lead user experiences during the COVID1-9 pandemic. Major comments: - It was unclear (in abstract, introduction, etc) whether the lead users were people experiencing covid, or happened to be lead users from before the pandemic and were being interviewed regarding care overall during the pandemic. Clarity on this topic would improve understanding of the paper. It wasn't until table 1 in results that it became clear that these participants hadn't necessarily experienced covid.- Overall, I think this paper is a valuable work documenting the experience of people during the COVID-19 pandemic. However, the tie to these people being 'lead users' is not very clear within the paper. I would either revise to make that more apparent, or consider reframing the paper (title, intro etc) to more generally describe the importance of understanding people's lived experiences during the pandemic.- It's not made clear to a reader how "healthcare being overburdened" results in "decreased opportunities for providing patient perspectives", with the "ordinary channels of patient involvement were no longer available" (first two sentences of discussion, I am summarizing here). Why does this matter? Is it more important to get care, or to have an opportunity to provide perspective? Was care ultimately reduced, or was this an unrealized fear during this time? Beyond stress and access to things like exercise classes, was health impacted in these lead users during this time period? This is important information, if it was collected within the study, to help contextualize the points I think the authors would like to make about the importance of then having patient involvement in the system.- Were any demographics collected from participants? If so, a table or a description within results would be beneficial. I see gender described (9 women, 1 man), but age and length of experience with the 'long-term' conditions would be helpful to contextualize the reported experiences.
--

	Minor comments:  - I found it challenging to contextualize the 'first phase' language, as I think that means different things to different readers. Even though the manuscript is focused in Sweden, I would suggest considering removing the 'first phase/wave' language and keeping the dates as a descriptor, or discussing "early" pandemic experiences without trying to categorize it as first phase. (e.g. in introduction it describes the first wave as starting in march 2020 but in the abstract setting it describes the first phase from June to September). It might seem pedantic but if it's discordant to other readers, it might be beneficial to remove the distraction. - In introduction third paragraph, who are the 'actors' being referred to? Consider other language to describe who you are talking about here? - It would be beneficial to include an example or expand on the first sentence of fourth paragraph in introduction regarding "the conditions for healthcare changed drastically". What did this mean in Sweden? - Design and recruitment – how were the 13 lead users who were invited identified? More data here (who did you target and why?) might be useful for future work building on this study. - Implications and future research – does this apply outside of Sweden, and if so, how might it be similar or different elsewhere?
--	---

REVIEWER	Brodeur, Magaly Universite de Sherbrooke, Département de médecine de famille et de médecine d'urgence
REVIEW RETURNED	03-Mar-2022

GENERAL COMMENTS	This study explores the experience of patient lead users during the COVID-19 pandemic. The topic is relevant and interesting. The text of the article is well written and easy to read. A major weakness of the article is the small number of participants in the qualitative interviews (N=10). This number does not allow for data saturation in qualitative research. The gender distribution also raises many questions (1 man / 9 women). We also know little about the participants in the study. A table presenting the profile of the participants (e.g., age, sex/gender, illness) would have been useful. The "methodology" section could be improved (e.g., present who did what in the "data collection" section as presented in the "data analysis" section). Did the patients lead users who were invited by email get reminded (if they did not respond)? Did the interviews were recorded in video format on Teams or only audio? The results section provides some interesting content. However, it should be improved. The analysis according to the themes "individual level" and "group level" is misleading. There is also a lack of information in many sections (e.g. p. 8 "At the time of the interviews, "participants" were unsure": How many participants?) The wording is often too general. The "strengths and limitations" section needs to be improved to better reflect the weaknesses of the study (small number of participants, majority of female participants, etc.).
--

	Finally, it is important to avoid "generalization" type wording in the conclusion. In summary, this study provides a relevant contribution to the literature. However, it needs to be reworked in order to be published.
--	--

VERSION 1 – AUTHOR RESPONSE

	Comments	Response
Reviewer 1		
1*	It was unclear (in abstract, introduction, etc) whether the lead users were people experiencing covid, or happened to be lead users from before the pandemic and were being interviewed regarding care overall during the pandemic. Clarity on this topic would improve understanding of the paper. It wasn't until table 1 in results that it became clear that these participants hadn't necessarily experienced covid.	We understand that it was unclear whether the lead users were people experiencing covid themselves or not. To make it clearer we have provided more information about the included participants in the abstract: "All participants were living with different long-term conditions and matched our definition of being patient lead users." In the introduction we addressed the case more specifically on page 14: "This caused tremendous uncertainty and worry for people who got infected by Covid-19 and other people in need of care". In the method section (Design and recruitment, page 6) we have clarified that Covid-19 infection was not a sampling criterion: "Personal experience of having been sick with Covid-19 was not an inclusion criterion".
2*	Overall, I think this paper is a valuable work documenting the experience of people during the COVID-19 pandemic. However, the tie to these people being 'lead users' is not very clear within the paper. I would either revise to make that more apparent, or consider reframing the paper (title, intro etc) to more generally describe the importance of understanding people's lived experiences during the pandemic.	We agree that the link to lead users was not evident enough and we are happy for the advice to revise the manuscript to make it more apparent. A section about the Swedish patient lead user network has been added in the introduction (page 4) to better place the study in the context of this existing, promising but still quite small patient initiative: "A patient lead user network was established around 2016-17 as a patient-driven initiative with the aim of supporting and stimulating the implementation of

		integrated healthcare systems from a patient (and informal caregiver) perspective. The network works across diagnoses in Sweden and internationally.”
3*	It's not made clear to a reader how “healthcare being overburdened” results in “decreased opportunities for providing patient perspectives”, with the “ordinary channels of patient involvement were no longer available” (first two sentences of discussion, I am summarizing here). Why does this matter? Is it more important to get care, or to have an opportunity to provide perspective? Was care ultimately reduced, or was this an unrealized fear during this time? Beyond stress and access to things like exercise classes, was health impacted in these lead users during this time period? This is important information, if it was collected within the study, to help contextualize the points I think the authors would like to make about the importance of then having patient involvement in the system	It is also unfortunate that it is not clear how healthcare being overburdened results in decreased opportunities for providing patient perspectives. According to the interviewees (as stated on page 9) , their own health did not change that much and it was not our main objective to explore. However, when talking about their experiences (of being a patient lead user in during the pandemic) the interviewees described a change when it came to the possibility of sharing their patient (lead user) perspective, activities that the participants of this study (as lead user patients) otherwise are used to. Our overall aim is to to deepen the understanding of patient lead users as a possible resource for the health system. In this study this was done by exploring their experiences during uncertain times such as a crisis, in this case the Covid-19 pandemic.
4*	Were any demographics collected from participants? If so, a table or a description within results would be beneficial. I see gender described (9 women, 1 man), but age and length of experience with the 'long-term' conditions would be helpful to contextualize the reported experiences.	We understand that more information about the participants would be beneficial. To better contextualise the reported experiences we changed the wordings when presenting the different long-term condition of the participants in the first sentence of the result (page 7): “The participant group entailed persons living with neurological, rheumatological, muscular, haematological, pulmonary, and metabolic long-term conditions as well as experiences of cancer, and paraplegy”. However, the population from which the sample is drawn, the Swedish patient lead user network, is small and we cannot add more details without compromising the confidentiality and anonymity of the participants. To better illustrate this a section about the Swedish patient lead

		user network has been added already in the introduction (page 4): “A patient lead user network was established around 2016-17 as a patient-driven initiative with the aim of supporting and stimulating the implementation of integrated healthcare systems from a patient (and informal caregiver) perspective. The network works across diagnoses in Sweden and internationally”.
5	I found it challenging to contextualize the ‘first phase’ language, as I think that means different things to different readers. Even though the manuscript is focused in Sweden, I would suggest considering removing the ‘first phase/wave’ language and keeping the dates as a descriptor, or discussing “early” pandemic experiences without trying to categorize it as first phase. (e.g. in introduction it describes the first wave as starting in march 2020 but in the abstract setting it describes the first phase from June to September). It might seem pedantic but if it’s discordant to other readers, it might be beneficial to remove the distraction.	Thank you for the comment! To make it clearer we have trougout the paper changed the wordings throughout the manuscript, from “first phase of the Covid-19 pandemic” to “early Covid-19 pandemic”.
6	In introduction third paragraph, who are the ‘actors’ being referred to? Consider other language to describe who you are talking about here?	To make it clearer who the “actors” are we have revised the sentence (page 4) and now more specifically write about people who are in need of care: “This caused tremendous uncertainty and worry for people who got infected by Covid-19 and other people in need of care”. We also added that it is patients “in general” who (according to a recent study) experienced anxiety etc: “a recent study showed that patients in general experienced anxiety and hesitation regarding care contacts”.
7	It would be beneficial to include an example or expand on the first sentence of fourth paragraph in introduction regarding “the conditions for healthcare changed drastically”. What did this mean in Sweden?	A paragraph has been added describing how the conditions for healthcare in Sweden changed drastically (page 5): “In the Swedish healthcare system, clinical wards for infectious diseases and intensive care units quickly needed to expand their capacity in terms of rooms, healthcare staff and medical equipment. In March 2020, the number of people infected with Covid-19 and needing inpatient care increased quickly in Sweden[15]. The capacity in clinics for infectious diseases and intensive care had to be expanded within weeks. Healthcare wards and staff usually

		working with infectious diseases focused only on providing care for persons with Covid-19, and staff from other wards were relocated to work on these dedicated Covid-19 wards[15]. Since healthcare resources needed to be relocated both within regions and nationally, the access to other types of healthcare was decreased. Retrospective studies of healthcare provided during the pandemic show that planned care was cancelled, care visits were converted from physical to digital appointments and people in need of care waited to book appointments in fear of burdening healthcare in this challenging situation[15].”
8	Design and recruitment – how were the 13 lead users who were invited identified? More data here (who did you target and why?) might be useful for future work building on this study.	We understand that the recruitment of the participants was unclear. To better describe the targeted population, from which the sample was drawn, a section about the Swedish patient lead user network has been added in the introduction (page 4): “A patient lead user network was established around 2016-17 as a patient-driven initiative with the aim of supporting and stimulating the implementation of integrated healthcare systems from a patient (and informal caregiver) perspective. The network works across diagnoses in Sweden and internationally”.
9	Implications and future research – does this apply outside of Sweden, and if so, how might it be similar or different elsewhere?	As with all qualitative research, context is of major importance to assess the transferability of the finding. In this particular study, transferability is affected by the different national regulations and levels of societal lockdown. But yes, the result might apply also outside of Sweden in countries where patients are normally invited to share their opinions. To clarify this we have added a sentence to the Implications and future research section (page 16): “We acknowledge that the societal response or action during the pandemic varied greatly in different countries, hence, this needs to be considered as an aspect of the transferability of our findings.”

Reviewer 2		
10	A major weakness of the article is the small number of participants in the qualitative interviews (N=10). This number does not allow for data saturation in qualitative research.	We agree that the small sample is a limitation and weakness of the study. In accordance with Varpio et al., we have chosen concepts other than saturation to secure quality in our data collection. To clarify our reasoning, we have added short description, and a reference, in the method section (page 5-6) "Following the arguments proposed by Varpio et al.[16], we aimed for a sample that is adequate (of sufficient size to allow transferability to other contexts), appropriate (with data that can answer the research questions) and aligned (with research questions and methodological choices)".
11	The gender distribution also raises many questions (1 man / 9 women). We also know little about the participants in the study. A table presenting the profile of the participants (e.g., age, sex/gender, illness) would have been useful.	We understand if the gender distribution raises questions. However, gender was not part of the sampling criteria and we do not analyse the collected data based on gender. We also understand that more information about the participants would be useful. To better contextualise the reported experiences we changed the wordings when presenting the different long-term condition of the participants in the first sentence of the result (page 7): "The participant group entailed persons living with neurological, rheumatological, muscular, haematological, pulmonary, and metabolic long-term conditions as well as experiences of cancer, and paraplegy". We have also added that all participants are adults: "Ten individuals (nine women, one man), all adults and identifying as patient lead users, were interviewed". However, the population from which the sample is drawn, the Swedish patient lead user network, is small and we cannot add more details without risking the anonymity of the participants. To better illustrate this a section about the Swedish patient lead user network has been added already in the introduction (page 4): "A patient lead user network was established around 2016-17 as a patient-driven initiative with the aim of

		supporting and stimulating the implementation of integrated healthcare systems from a patient (and informal caregiver) perspective. The network works across diagnoses in Sweden and internationally."
12	The "methodology" section could be improved (e.g., present who did what in the "data collection" section as presented in the "data analysis" section).	Thank you for the comment! In the revised manuscript we state that the the interviews were conducted by a female research assistant, but this was moved to the second paragraph under data collection, more specifically about the interviews (page 6): "Individual single interviews were conducted by a female research assistant trained in qualitative methods and interview techniques". We have also clarified who sent out the invitations (page 5): "E-mail invitations were sent by two patient lead users that were co-researchers in the project" and elaborated the text about the development of the interview guide (page 6): "The first draft of the interview guide was created by TS, SR and HH and later jointly developed by the research team".
13	Did the patients lead users who were invited by email get reminded (if they did not respond)? Did the interviews were recorded in video format on Teams or only audio?	The recruitment procedure has been clarified (page 6) with information about reminders: "Reminders were sent if no response was received. In total, 13 patient lead users were invited, and reminders were sent to those who did not respond. At the end 10 agreed to be interviewed. Personal experience of having been sick with Covid-19 was not an inclusion criterion." and recordings: "The interviews, conducted using Microsoft Teams, lasted 35–86 minutes (average 60 minutes), and were audio recorded (no video) and transcribed verbatim".
14	The results section provides some interesting content. However, it should be improved. The analysis according to the themes "individual level" and "group level" is misleading.	Thank you for highlighting that the themes individual and group level are misleading. Unfortunately we are not certain what is misleading, but hope that we have made the distinction a bit clearer by changing the name of the second theme from "group level" to "system level". We are happy to revise more if this still is misleading (e.g., delete

		the two levels from the analysis and directly present the six sub-themes). The change from “group level” to “system level” has been made in the abstract (page 2), result section first paragraph (page 7), table 1 (page 8) and page 10.
15	There is also a lack of information in many sections (e.g. p. 8 "At the time of the interviews, "participants" were unsure": How many participants?) The wording is often too general.	It is unfortunate that the result section gives the impression information is missing in many sections and that the wordings in too general. The basis of our wording is based on research from e.g. Monrouxe et al. (When I say ... quantification in qualitative research) arguing that numbers should not be reported if not explored for the entire sample, percentages should not be used for smaller samples (n<50) and semi-quantification should not be used without justification.
16	The "strengths and limitations" section needs to be improved to better reflect the weaknesses of the study (small number of participants, majority of female participants, etc.).	We agree that the Strengths and Limitations section had to be improved to better reflect the weaknesses of the study. To clarify the limitations there is now a separate bullet point about about the relatively small sample and its possible implication (page 3): The relatively small sample might affect the reliability of the study. In the Strengths and weaknesses section (page 15) it is also added that “A sample with a majority of female participants could affect the generalisability of the result but gender was not a sampling criterion in this study”.
17	Finally, it is important to avoid "generalization" type wording in the conclusion.	We agree that it is important to not use “generalization” type wordings in the conclusion. To avoid this we have added a few words to the Conclusions section (page 16), for exemple “a number of (patient lead users)”, “responents experienced that” and “were considered (unable to)”.
Editor		
18*	Move the Strengths and Limitations section so that it follows the Abstract	The Strengths and Limitations sections has been moved and now follows directly on the Abstract (page 3).

19 *	Revise the ‘Strengths and limitations’ section (after the abstract). This section should contain up to five short bullet points, no longer than one sentence each, that relate specifically to the methods. The novelty, aims, results or expected impact of the study should not be summarised here.	The Strengths and limitations section has been revised (page 3) and there is now a separate bullet point about the relatively small sample:  • The qualitative and inductive approach contribute to an in-depth and rich exploration of patient lead users’ experiences during the Covid-19 pandemic. • A sample consisting of respondents with different chronic conditions and life situations. • A patient lead user was part of the research team, which can contribute to improving the relevance of research to patients. • The relatively small sample might affect the reliability of the study.
20	Provide an Ethical statement in your main document under heading of "Ethics statement" and embed it just before reference list. The statement in the ScholarOne system and main document should be the same.	An Ethical statement is now provided also in the main document under the new heading “Ethics statement” and embedded just before the reference list (page 17): “Swedish Ethical Review Authority, 2019-03849 with amendment 2020-01741” .
21	Provide the following statements below and embed it just before reference list. a. Contributorship statement Please provide contributorship statement by indicating each author's contribution to the paper according to the ICMJE guidelines for authorship. This should be stating how each author contributed to the article. It should discuss on the planning, conduct and reporting of the work in your paper. You may also consider the conception and design, acquisition of data or analysis and interpretation of data, etc. The statement in the ScholarOne system and main document should be the same. If anyone currently listed as an author does not fulfill all three of these then they should be moved to the acknowledgment section. b. Competing Interest Ensure that the Competing Interest in ScholarOne system and main document should be the same. c. Funding Word your funding statement as follows. Either: "This work was supported by [name of funder] grant number [xxx]?" or "This research received no specific grant from any funding	A contributorship statement was written in the manuscript under the heading “Author contribution”, now changed to “Contributorship” and slightly revised (page 17): “HH and SR initiated the study. TS led the data collection. SR identified and invited participants. TS and HJ analysed the data, and TS developed preliminary themes that were reviewed by HJ. HH drafted the background, TS drafted the result, and HJ drafted the method and discussion sections of the manuscript. All authors revised, edited, and approved the final manuscript. MR was responsible for the overall content as the guarantor. HJ, as the corresponding author, attests that all listed authors meet authorship criteria and that no others meeting the criteria have been omitted”. The following headings and texts were missing in the main document but are now added (page 17), just before the Reference list, and correspond to the ScholarOne system:

	agency in the public, commercial or not-for-profit sectors?. The statement in the ScholarOne system and main document should be the same. d. Data sharing Provide a data availability statement in your main document. Specify what unpublished data are available and where it can be accessed. If there are none then you can simply state "No additional data available". The statement in the ScholarOne system and main document should be the same.	“Competing interest The authors have nothing to declare. Funding This work was supported by the Swedish Research Council for Health, Working Life and Welfare (FORTE) grant number 2018-01472. Data sharing No additional data available”.
22	Authors must include a statement in the methods section of the manuscript under the sub-heading 'Patient and Public Involvement'. This should provide a brief response to the following questions:  - How was the development of the research question and outcome measures informed by patients’ priorities, experience, and preferences? - How did you involve patients in the design of this study? - Were patients involved in the recruitment to and conduct of the study? - How will the results be disseminated to study participants? - For randomised controlled trials, was the burden of the intervention assessed by patients themselves? - Patient advisers should also be thanked in the contributorship statement/acknowledgements 	The sub-heading Patient and Public involvement and corresponding text was missing in manuscript but is now added, in the methods section (page 7), and correspond to the ScholarOne system: “If a study aims to improve the relevance of research to patients, this must also be reflected in the research process used.[19] Therefore, the research team for this study included authors with formal experience as researchers, healthcare workers, and with lived experience as patient lead users. A patient lead user with lived experience of long-term disease has co-authored this article, been part of the research team and has contributed in all parts of the research process”. Furthermore, the results will be disseminated through the patient lead user network.
23	‘Strengths and limitations of this study’ should be placed after the abstract.	The Strengths and Limitations sections has been moved and now follows directly on the Abstract (page 3).

VERSION 2 – REVIEW

REVIEWER	Lewis, Dana OpenAPS, Seattle
REVIEW RETURNED	10-May-2022

GENERAL COMMENTS	Thank you for revising this manuscript and addressing the previous comments around confusion of lead user, recruitment methods, etc. The revised version is much improved for reader comprehension.
---